# Priapism in a Patient with Rectal Adenocarcinoma

**DOI:** 10.3390/diseases11010034

**Published:** 2023-02-17

**Authors:** Navdeep Dehar, Justin Tong, Zain Siddiqui, Michael Leveridge, Anna Tomiak

**Affiliations:** 1Department of Oncology, Queen’s University School of Medicine, Kingston, ON K7L 2V7, Canada; 2Cancer Centre of South Eastern Ontario, Kingston Health Sciences Centre, Kingston, ON K7L 2V7, Canada; 3Department of Urology, Queen’s University School of Medicine, Kingston, ON K7L 2V7, Canada

**Keywords:** priapism, metastatic, rectal cancer, colorectal cancer

## Abstract

Background: Priapism is a very rare complication of malignancy and is usually accompanied by locally advanced or widely metastatic disease. We describe a case of priapism arising in a 46-year-old male with localised rectal cancer that was responding to therapy. Case presentation: This patient had just completed two weeks of neoadjuvant, long-course chemoradiation when he presented with persistent painful penile erection. Assessment and diagnosis were delayed for more than 60 h, and although a cause could not be determined from imaging, a near complete radiological response of the primary rectal cancer was seen. His symptoms were refractory to urologic intervention and were associated with extreme psychological distress. He re-presented shortly thereafter with extensively metastatic disease in the lungs, liver, pelvis, scrotum, and penis; additionally, multiple venous thromboses were identified, including in the dorsal penile veins. His priapism was not reversible and was associated with a considerable symptom burden for the remainder of his life. His malignancy did not respond to first-line palliative chemotherapy or radiation, and his clinical course was further complicated by obstructive nephropathy, ileus, and genital skin breakdown with a suspected infection. We initiated comfort measures, and he ultimately died in hospital less than five months after his initial presentation. Conclusion: Priapism in cancer is usually related to tumour infiltration of the penis and corporal bodies resulting in poor venous and lymphatic drainage. The management is palliative and can include chemotherapy, radiation, surgical shunting, and potentially penectomy; however, conservative penis-sparing therapy may be reasonable in patients with limited life expectancy.

## 1. Introduction

Malignant priapism is a rare phenomenon mostly associated with penile metastasis and has a poor prognosis [1,2,3,4,5,6]. The earliest description of penile metastasis dates back to 1870, with the first extensive review published in 1961; between then and 2018, approximately 460 cases were reported, of which 19% originated from a colorectal primary site [4]. Meanwhile, priapism in the absence of an advanced disease is even more unusual. We present the case of a man with rectal adenocarcinoma who developed priapism during combined modality chemoradiation in the neoadjuvant setting. The patient provided informed consent for the use of his personal health information in this case report, and approval was obtained from our local research ethics board.

## 2. Case Presentation

This 46-year-old, previously healthy Caucasian male presented with a history of chronic, intermittent, painless rectal bleeding, and a change in his bowel habits that had occurred for 3–4 years. His past medical history was significant for a congenital imperforate anus, which had been surgically corrected, substance abuse, depression, and recurrent urinary tract yeast infections, causing urethral discomfort. He was diagnosed with a cT2N1M0 adenocarcinoma of the distal rectum and began long-course chemoradiation therapy with capecitabine, with the intent of downstaging the tumour prior to surgical resection.

On the first day of treatment, the patient reported a burning perineal pain radiating from the penis. As the pain persisted, he was assessed by urology staff on day 13. The urological examination at that time was normal with no skin changes or abnormalities noted in the penis. The urine cultures were negative. The pain was thought to be radiating from the rectal primary tumour, unrelated to any primary penile pathology, and was managed conservatively.

On day 18 of chemoradiation, the patient’s penile pain became more severe, and his penis became persistently erect. He presented to the hospital emergency room but left the emergency department before being seen. He reported his symptoms to the oncology service two days later, at which point, approximately 60 h had elapsed without treatment. The physical examination was consistent with priapism with a swollen, rigid, erythematous penile shaft, with blue discoloration at the base of the glans penis (Figure 1). He complained of significant local pain, the inability to urinate, and difficulty ambulating caused by the degree of swelling. A penile blood ABG showed pH 7.3, pO_2_ 56, and pCO_2_ 50, consistent with ischemic priapism. He was admitted to the urology department for further management. Phenylephrine injections failed to resolve the priapism; as such, he was taken to the operating room, where he underwent corporal aspiration, T shunting, and a Hegar dilation of the corporal bodies. Repeated MRI scans of the pelvis showed significant corporal body fibrosis, but there was no evidence of tumour invasion or metastasis (Figure 2). The patient initially endorsed some symptomatic benefits, but within the next few days, he again reported worsening symptoms, including a persistent erection associated with extreme pain, anxiety, and distress; in addition, the swelling had extended to the scrotum. The urology staff recommended a conservative symptomatic treatment due to the recency of his surgical interventions. Following a Tumour Board discussion, a decision was made to discontinue further chemoradiation due to a combination of factors including the positive response seen on the MRI scans, the possibility that the priapism was an adverse treatment effect, and to facilitate symptomatic relief. At that point, he had received 14 out of the 25 planned radiation fractions to the pelvis (2520 cGy delivered) and 0 out of 5 fractions to the rectum, alongside concomitant chemotherapy. The plan was for the patient to proceed directly to surgery after additional wait and downstaging.

This plan required alteration, as the patient presented again to the hospital three weeks later with worsening symptoms from persistent priapism. This time he also presented with dyspnea and was found to have undergone rapid disease progression, with imaging evidence of new bilateral lung metastases, hepatic metastases, pelvic and inguinal adenopathy, and pubic bone lytic lesions. He was also found to have developed multiple venous thromboemboli with a high clot burden, including bilateral subsegmental pulmonary emboli and thromboses of the right external iliac, left greater saphenous, and bilateral dorsal penile veins (Figure 3, Figure 4, Figure 5, Figure 6, Figure 7, Figure 8 and Figure 9). In addition, there was increased spiculated soft tissue swelling along the dorsal penile shaft and extensive nodularity in the scrotum and inguinal lymph nodes, suggesting tumour invasion of the reproductive organs (Figure 4, Figure 5 and Figure 6). During the physical examination, the penis, scrotum, and pre-pubic areas were dramatically enlarged and firm, with numerous cutaneous lesions consistent with replacement with a tumour. Urology staff were consulted, and they suggested that the disease was not resectable.

He began dose-reduced FOLFIRI chemotherapy with palliative intent, with the best response being progression after two cycles. His course was also complicated by rapid atrial flutter and hypotension requiring pressors, which were attributed to the progression of the thromboembolic disease, requiring an increase in the anticoagulation dose. Throughout these events, the patient’s principal complaint remained the persistent priapism and associated pain and psychological distress. He was connected to both the palliative care and psychiatry departments and continued follow-up with the urology department, who inserted a suprapubic catheter to reduce the urinary retention. The patient received palliative radiation (20 Gy in 5 fractions) to the groin with the goal of controlling his groin lymphadenopathy, which was thought to be contributing to his priapism (Figure 10A–D). There was a plan to initiate second-line FOLFOX shortly after completing the radiation treatment.

Despite palliative chemotherapy and palliative radiation, the patient’s clinical status continued to deteriorate. He developed worsening lethargy, poor intake, renal failure, and open penile and scrotal sores with purulent drainage, indicative of a superimposed infection, and he was re-admitted for further management. Imaging showed the significant further progression of metastatic disease (Figure 7, Figure 8 and Figure 9), especially in the pelvis, with new bilateral hydroureteronephrosis and evidence of ileus. He was deemed to no longer be a candidate for further radiation or systemic therapy because of his declining clinical condition. The urology staff again recommended against the resection of the scrotum and penis. He elected to pursue comfort measures only and died in hospital.

## 3. Discussion

Colorectal cancer is the third most common cause of cancer-related mortality worldwide and the fourth most common malignancy presenting with metastasis, with 20% of cases presenting with a metastatic disease [1,2]. The most common sites of metastasis are the regional lymph nodes, liver, lungs, and peritoneum; penile involvement, however, is very rare [3,4]. Indeed, metastasis to the penile tissue is uncommon across all tumour types, and this is especially true in cancers of non-urogenital origin (5,6). While up to 70% of penile metastases arise from urogenital tumours, 20% originate in the gastrointestinal tract [7].

Priapism is defined as a persistent, often painful, penile erection, which is unrelated to sexual stimulation and lasts more than four hours [8]. Though ischemia from vascular disease or intracavernosal injection accounts for approximately 95% of all cases, there remain cases in which the cause is not clear; in such cases, especially if significant risk factors are present, malignancy should be suspected [9]. Penile metastasis can cause either ischemic priapism, by obstructing venous drainage from the corpus cavernosa, or high-flow priapism, by increasing arterial flow to the corpora via arterial fistulisation [10]. Penile metastasis from rectal carcinoma, in particular, has been previously reported by Yildrim et al. and Marcu et al. [8,9,11] in locally advanced and extensively metastatic disease.

This patient’s presentation was very unusual in comparison to those of previously reported cases for several reasons. Firstly, his priapism first arose with minimal residual disease, with no penile or perineal disease being evident, and after a near-complete radiological response to neoadjuvant therapy, whereas previous cases have occurred in the setting of locally advanced or metastatic disease. Secondly, the behaviour of his rectal cancer was unusually aggressive, with rapid development of widespread metastatic disease and local extension to uncommon sites of metastases. His re-presentation was consistent with penile tissue replacement by a tumour. Thirdly, there were multiple factors at play that make the determination of a clear singular aetiology difficult: delayed diagnosis, a high burden of thrombosis, radiation-induced changes in the surrounding areas, violation of the anatomy by urologic intervention, as well as clear disease progression in the end stage. Although the aetiology of the patient’s ischemic priapism was never clearly established, a number of different causes were considered. These included the possibility of micro-metastases to the corpora-cavernosal tissue that were missed on imaging, but were occluding cavernosal outflow, causing priapism and an associated prothrombotic state. The possibility of a primary thromboembolic event was also considered, though this was considered to be less likely, with no previous case reports in the literature. Other potential causes include lymphangitic spread and obstruction, which may help explain the skin turgor. Any or all of these processes could also have been exacerbated by local microangiopathic effects of the radiation. The tense and expanded corpus spongiosum is uncommon in typical priapism, which affects the corpora cavernosa, and suggests a more proximal or a more systemic cause such as micrometastases or a lymphangitic spread. The rapid spread of the disease, including penile and scrotal cutaneous metastases, lends credence to a process defying imaging resolution, but with overt clinical manifestations.

## 4. Conclusions

Priapism is a rare complication of malignancy, and as reported here, it can occur even without an advanced disease or definite penile metastases. It can cause major detriment to a patient’s quality of life and may portend a poor prognosis, although this particular patient’s trajectory may have been affected by a delay in diagnosis. Early assessments and interventions, with the hope of reversing and preventing significant and debilitating symptoms that could otherwise become a major source of morbidity for the remainder of life, should be prioritised in future instances.

## Figures and Tables

**Figure 1 diseases-11-00034-f001:**
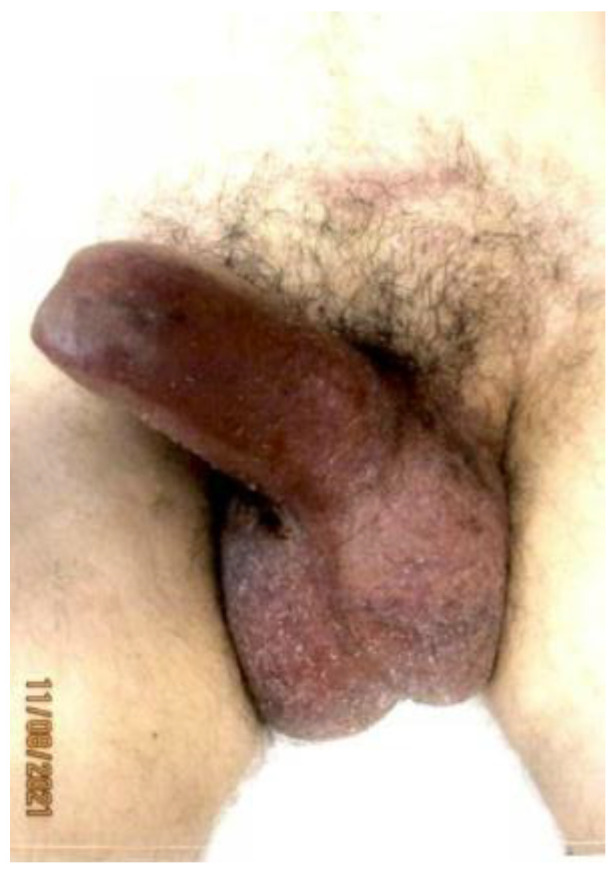
Priapism at two weeks post-neoadjuvant chemoradiation.

**Figure 2 diseases-11-00034-f002:**
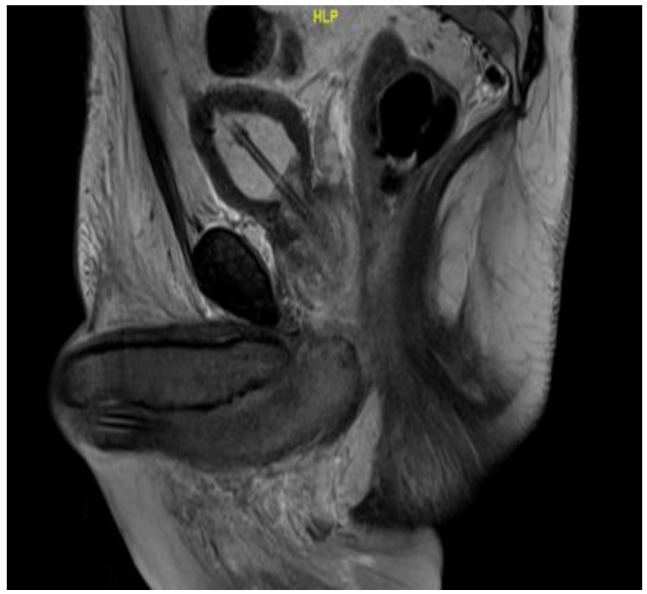
Sagittal MRI T2 scan showing enlarged bulbospongiosus.

**Figure 3 diseases-11-00034-f003:**
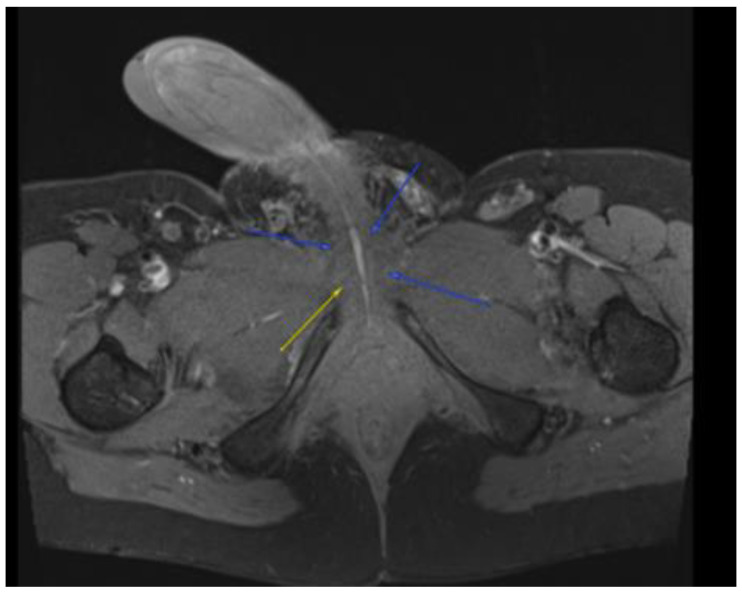
MRI in the axial view T1 TSE FS PRE showing hyperintense linear non-enhancing structure consistent with ultrasound finding of thrombosed dorsal penile veins.

**Figure 4 diseases-11-00034-f004:**
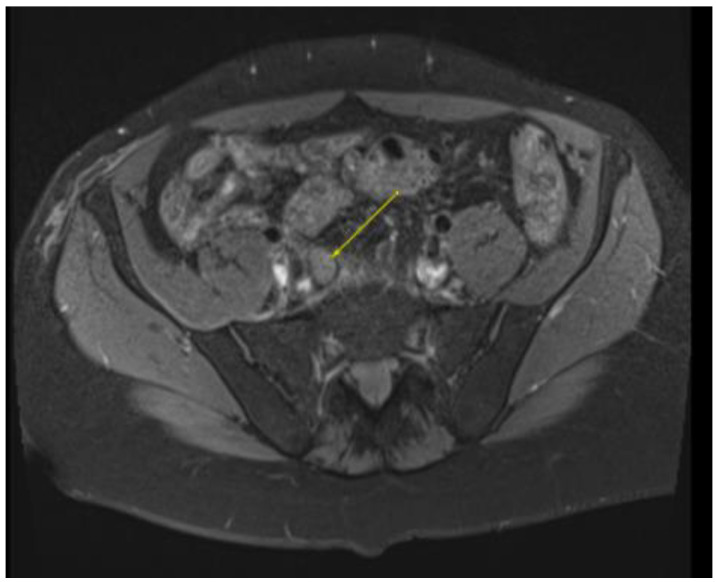
MRI in the axial view T1 TSE FS pre-showing 11 mm R common iliac lymph node.

**Figure 5 diseases-11-00034-f005:**
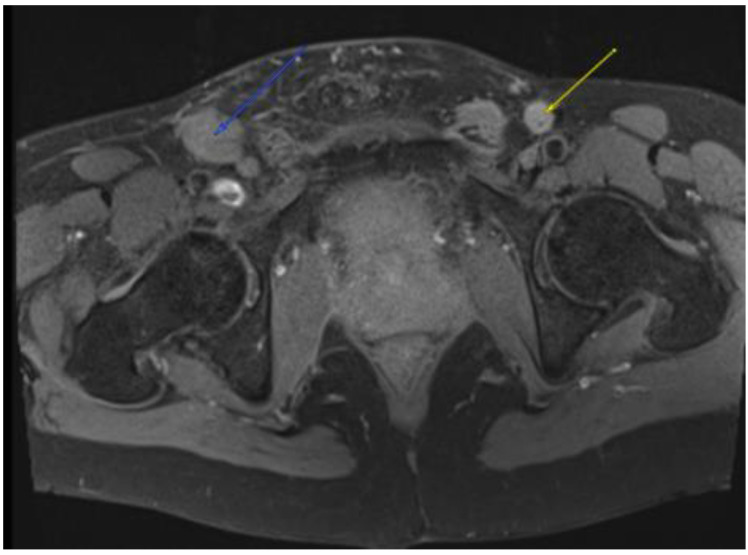
MRI in the axial view T1 TSE FS pre-showing 22 × 30 mm bilateral inguinal lymph node.

**Figure 6 diseases-11-00034-f006:**
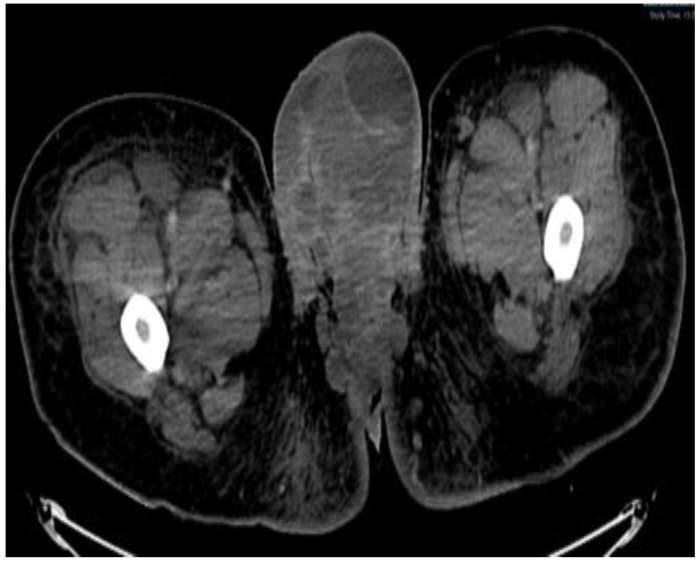
CT scan in the axial view, abdominal window, showing scrotal mass extending through the anus with involvement of ischial tuberosity.

**Figure 7 diseases-11-00034-f007:**
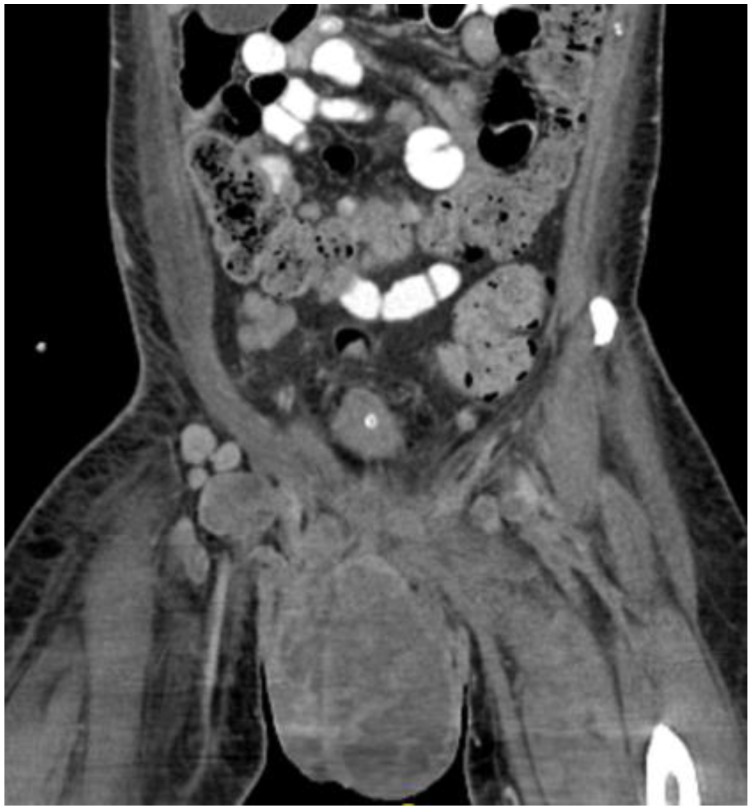
CT in the coronal view of scrotal mass in the right inguinal lymph node.

**Figure 8 diseases-11-00034-f008:**
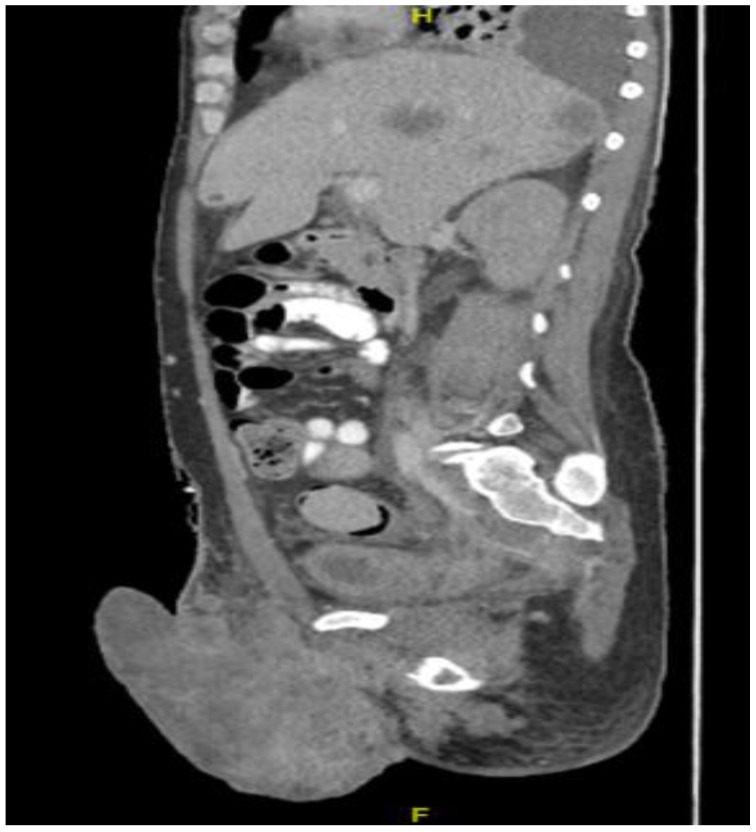
CT in the sagittal view, abdominal window, showing scrotal mass extending to erect penis.

**Figure 9 diseases-11-00034-f009:**
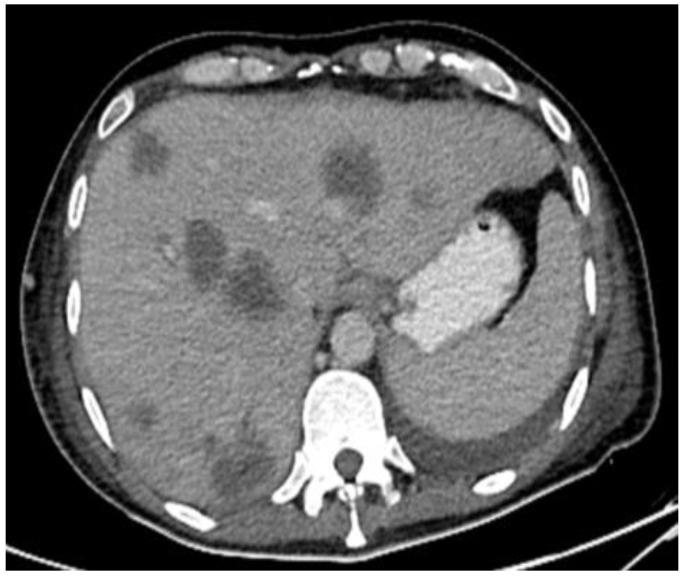
CT in the axial view of an abdominal window showing liver metastasis.

**Figure 10 diseases-11-00034-f010:**
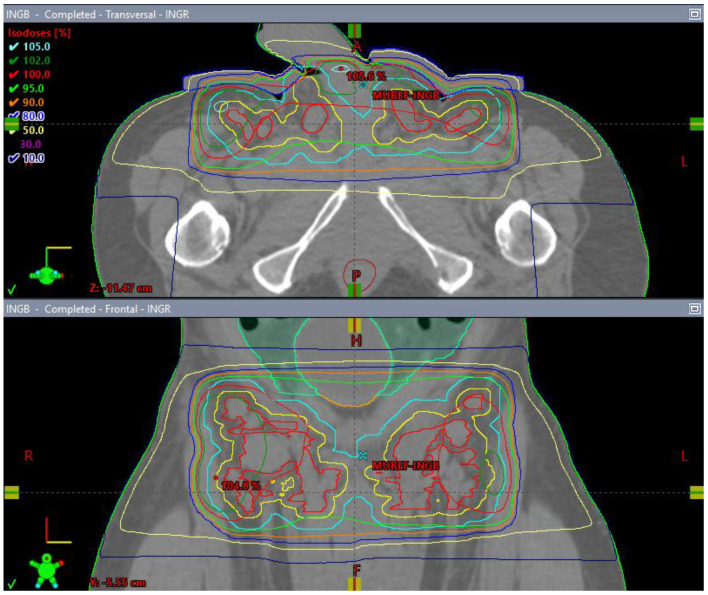
3 Field (2 laterals (**A**,**B**) and 1 posterior (**C**) field) palliative radiation plan with 20 Gy/5 fractions to the inguinal nodes with a 1 cm bolus (**D**).

## Data Availability

Not applicable.

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
