# Peer review of "Priapism in a Patient with Rectal Adenocarcinoma"

_diseases, 2023, doi:10.3390/diseases11010034_

Round 1

Reviewer 1 Report

Very clear and interesting case presentation. Very good high quality images.  No major issues; I have only one minor recommendation :

Your conclusion is very 'abstract', partly repeats senetnces from introduction. This should be more based on your case.

Author Response

Thank you for the comments.

  • Reviewer comment: Your conclusion is very 'abstract', partly repeats sentences from introduction. This should be more based on your case.
  • Author response: The conclusion has been revised as per the comment.

Reviewer 2 Report

This paper is clearly written and well organized. The introduction and background are reasonable given the premise of the paper. The authors explain very well in this paper about Priapism in a patient with rectal adenocarcinoma. I am recommending to publishing this paper without any modification.

Author Response

This paper is clearly written and well organized. The introduction and background are reasonable given the premise of the paper. The authors explain very well in this paper about Priapism in a patient with rectal adenocarcinoma. I am recommending to publishing this paper without any modification.

Response:

We thank the reviewer for their time and feedback.

Reviewer 3 Report

This paper is another decription of the case of priapism in a colorectal cancer-patient. The case report is profoundly described, contains a detailed imaging documentation, and is well supported with available references.

Although overall scientific soundness is rather low, the manuscript indicates a rare erectile complication which remained a devastating burden for the ill person till the end of their life. 

This case had a particular presentation, as the condition developed after near-complete radiological response to chemotherapy, not in an advanced setting.

The paper re-summarizes the actual information on etiology of priapism in cancer diseases, and may be of help when deciding on clinical approach in patients with malignant priapism.

Author Response

Comments and Suggestions for Authors

This paper is another decription of the case of priapism in a colorectal cancer-patient. The case report is profoundly described, contains a detailed imaging documentation, and is well supported with available references.

Although overall scientific soundness is rather low, the manuscript indicates a rare erectile complication which remained a devastating burden for the ill person till the end of their life. 

This case had a particular presentation, as the condition developed after near-complete radiological response to chemotherapy, not in an advanced setting.

The paper re-summarizes the actual information on etiology of priapism in cancer diseases, and may be of help when deciding on clinical approach in patients with malignant priapism.

Response: 

We thank the reviewer for their time and feedback.

Reviewer 4 Report

Minor English polishing equired

Uneven font seen in text.

Abstract too long , with case presentation itself case report and write a single sentence about conclusion.

Introduction

Where are references 1-3, directly starts from reference 4.

Is unintroductory , cite some evidence based references.

Case Presentation

If general reader reads case report , he will understand that cTN1M0

What was type of pain ?

What was result of urine examination ?

Do you had any Doppler study of penis ?

You have not excluded Peyrones disease ?

You are conluding ..which was thought 119 to be contributing to his priapism which was thought .. in clinical science , there is no thought, only likely or suggested

You are not systematic in reaching cause of your priaprism

Then in discussion cause changes

You could have stopped erection just by giving low dose of estrogen

The cause of priapism in this case is mutiple, go deep insight in your case.

Author Response

Response:

We thank the reviewer for their feedback and suggestions. Our responses are as follows:

-The manuscript has been proofread for English readability.

-Uneven font in the manuscript has been corrected.

-We note that the abstract meets MDPI requirements in its current form. The abstract's current length is based on author preference.

-Introduction

-References 1-3 are now cited in the body of the text.

-We feel the introduction is reasonable for the premise of the paper, as have other reviewers. Evidence-based references have already been cited as much as possible, including systematic reviews.

-Case Presentation

-The patient's pain was reported as burning in nature, initially originating in the perineum with radiation up the penile shaft, and later more concentrated in the penis.

-Urine examination was unrevealing, with no pyuria or proteinuria, and negative urine cultures.

-A Doppler US of penis was conducted at the time of his presentation with venous thromboemboli and showed bilateral dorsal penile vein thromboses.

-The patient was assessed by the Urology team on multiple occasions. His symptoms were not consistent with Peyronie's disease.

-We feel the use of "thought" in medical writing is a reasonable stylistic choice and up to author preference, because thought is an essential component of the practice of medicine.

-One of the challenging features of this case was that the exact cause of the priapism was never definitively determined. All that was known for certain was that he had an ischemic priapism based on the blood gas, without any pharmacologic cause. A number of different causes were considered, such as the possibility of micro-metastases to the corpora-cavernosal tissue that were missed on imaging but were occluding cavernosal outflow, causing priapism and an associated prothrombotic state. The possibility of a primary thromboembolic event was also considered, though this was considered to be less likely with no previous case reports in the literature. Other potential causes include lymphangitic spread and obstruction, which may help explain the skin turgor. Any or all of these processes could also have been exacerbated by local microangiopathic effects of the radiation.

-The exquisite pain and sensitivity are less common, suggesting effects at the cutaneous level. The tense and expanded corpus spongiosum is uncommon in typical priapism, which affects the corpora cavernosa, and suggests a more proximal or a more systemic cause like micrometastases or lymphangitic spread. His explosion of disease, including penile and scrotal cutaneous metastases, lends credence to a process defying imaging resolution but with overt clinical manifestations (priapism and significant pain).

Round 2

Reviewer 4 Report

Accept